# Interventions to reduce social isolation and loneliness among minority ethnic populations in OECD countries: A scoping review

Emmanuel Sunday Nwofe[ID]*, Amirah Akhtar[ID], Sahdia Parveen, Karen Windle

Centre for Applied Dementia Studies, University of Bradford, Bradford, United Kingdom

* e.s.nwofe@bradford.ac.uk

**Data Availability Statement:** All relevant data are within the paper and its Supporting Information files.

## Abstract

### Background

Social isolation and loneliness (SIL) are complex issues that impact mental and physical wellbeing and are significant public health concerns. People from minority ethnic backgrounds living in Organisation for Economic Cooperation and Development (OECD) member states may be particularly vulnerable to experiencing SIL. This is due to various challenges associated with life in foreign countries, including cultural differences, settlement issues, low incomes, and discrimination. While many interventions have been developed to address SIL in the general population, there is little information about interventions designed for minority ethnic populations in OECD countries. Our study aimed to 1) Investigate existing interventions for minority ethnic communities in OECD countries; 2) Assess how these interventions are conceptualised to increase awareness of SIL risks on health factors.3) Explore culturally sensitive approaches in these interventions, and 4) Identify the most effective interventions in reducing SIL in minority ethnic populations.

### Methods and findings

We searched Medline, APA PsycINFO, Psychology and Behavioural Sciences Collection, CINAHL, Web of Science, and Scopus from their inception to September 19th, 2023, and registered the scoping review at https://osf.io/fnrvc. Our search yielded 10,479 results, of which 12 studies were included: five RCTs, six non-randomized quasi-experimental studies, and one qualitative study. Interventions were grouped into four main categories: social facilitation, befriending, leisure and skills development, and health education programmes. While only a few interventions targeted minority ethnic populations specifically, our findings highlight the potential of culturally sensitive interventions in reducing social isolation and loneliness among minority ethnic communities in OECD countries. However, given the type and extent of evidence, it is still unclear which interventions are superior in reducing SIL in minority ethnic populations in OECD countries. Further research is necessary to understand which activities may be most effective for which communities. Such interventions should be designed and tailored to account for the broader risk implications of SIL to raise awareness of the population's peculiar health risk profile.

**Funding:** The author(s) received no specific funding for this work.

**Competing interests:** The authors have declared that no competing interests exist.

## Discussion

Interventions designed to address SIL among minority ethnic groups in OECD member states are scarce and have not been designed to account for the health risks profile of the population. Integrated research designs involving groups linked with minority ethnic populations are needed to link individual, community, and societal factors alongside population risk profiles for increased recognition of SIL as an important health factor.

## Introduction

Social isolation and loneliness have long been a public health concern due to their negative impacts on individual health and well-being, especially considering their links with factors that are unequally distributed in society [1]. They are now receiving additional attention following their inclusion among dementia risk factors. Still, social isolation and loneliness are often neglected social determinants for people of all ages, including older adults. Recently, the World Health Organisation established a Commission on Social Connection to optimise the recognition and resourcing of SIL as a priority public health problem [2].

Social isolation refers to the objective lack of social contact with other people or infrequent social connections. In contrast, loneliness is defined as a subjective feeling caused by a discrepancy between one's desire for connections and the actual degree of connectedness [3, 4].

A recent systematic review and meta-analysis involving 1.3 million participants across 20 countries found a significant 35% higher hazard of all-cause mortality among those who are socially isolated [5]. Social isolation and loneliness also contribute to depression [6] and have been associated with myriads of other physical conditions, including cardiovascular diseases, diabetes, and stroke [7, 8]. There is compelling evidence linking social isolation and loneliness with worse neurocognitive markers of Alzheimer's disease and related dementias [9–11].

The prevailing assumption of the theory of loneliness is that experiencing loneliness and feeling disconnected from others is a common consequence of marginalisation [12]. It is suggested that social isolation and loneliness will be higher among members of the marginalised groups, including those from minority ethnic backgrounds in OECD countries. The expectations in the quantity/ quality of social relationships are culturally located [13], and living in a foreign culture can set a new risk for loneliness [14, 15]. This could be a result of cultural clashes or differential levels of exposure to loneliness vulnerability [16].

Previous studies have indicated that people who have migrated to another country may be more susceptible to experiencing loneliness than their native-born counterparts or those in their home country [17, 18]. A review of the prevalence and trends of social isolation and loneliness in Scotland associated increased risk of social isolation and loneliness with being from a minority ethnic background [19]. This may be due to weakened social norms and can vary based on exposure to loneliness risk factors such as poor health, low income, deprived neighbourhood, and language barriers [20–22]. The British Red Cross found that people from Black, Asian, or minority ethnic backgrounds are 38% more likely to feel lonely compared to 28% of White people; those who have experienced discrimination at work or in their local neighbourhood reported feeling lonely always or often, compared to those who have not [23]. These factors, in addition to being from a minority ethnic community, increase feelings of powerlessness and unfulfilled expectations that often exacerbate mental health vulnerability [24]. Yet, social isolation and loneliness are often under-recognised public health problems among minority ethnic communities. Amidst a plethora of studies aiming to address social

isolation and loneliness in the general population, not much is known about interventions aimed to reduce social isolation and loneliness among minority ethnic populations in OECD countries. Available review articles either focused on older ethnic minority experiences [25, 26] for specific ethnicity [27], in a single country context [28] or explored interventions for the elderly [29] beyond ethnic and geographical limitations [30].

We aimed to investigate 1) What interventions addressing social isolation and loneliness have been designed for minority ethnic communities in OECD countries from early to later life (18 years and older)? 2) How well do the hypothesised mechanisms of action link interventions to raising awareness of SIL risks on health factors? 3) Which culturally sensitive approaches were used? 4) Which interventions are most effective in reducing SIL amongst ME communities?

## Methods

To identify and describe the nature of evidence and interventions designed to address social isolation and/or loneliness in minority ethnic populations, scoping review procedures were applied to systematically search, screen, extract and chart this broad body of literature. We searched Medline, APA PsycINFO, Psychology and Behavioural Sciences Collection, CINAHL, Web of Science and Scopus from inception to September 19th, 2023, for peer-reviewed primary studies. At least two authors screened studies and extracted data using the Joanna Briggs Institute (JBI) scoping review checklist. Since our aim was not to assess intervention efficacy, a quality appraisal was not conducted as typical in the scoping review methodology [31]. We used the population, concept, and context (PCC) [32] approach when developing the research questions and search strategy, where the population refers to the minority ethnicity, the concept of social isolation and loneliness and the context of community setting in OECD countries. A protocol containing the rationale, objectives, research questions and detailed methodology was prospectively registered in Open Science Framework (https://osf.io/fnrvc) in September 2023. We reported according to the Preferred Reporting Items for Systematic Reviews and Meta-Analyses Extension for Scoping Reviews (PRISMA-ScR) guidelines **S1 Table** [31].

### Eligibility criteria

Articles were included if: (1) they focused on ethnicity other than white backgrounds or compare people from minority ethnic populations with white backgrounds, 2) included at least one of the minority ethnic groups: Black African/Afro-Caribbean and/or South Asian, 3) measured loneliness and/or social isolation including constructs of social connectedness such as social support, social engagement, social participation, or relations, 4) be conducted in any of the 38 OECD member states, 5) be written in English language and 6) involved participants in community settings. We excluded studies if: 1) conducted in hospital settings (e.g., care homes and nursing homes), 2) included a sample below 18 years of age, (3) delivered pharmacological interventions or reported interventions not yet implemented, (4) were narrative summaries, commentaries, opinion pieces, reviews and/or study/review protocols.

### Definition

We used community settings to describe interventions delivered in the community, including community centres or playgrounds, participants' homes, or other locations, excluding hospital and/or residential care settings where participants receive 24-hour care. Interventions could be face-to-face, virtual, telephone, or video calls. Care facilities, such as care homes, already

provide physical and psychological support to residents in the setting, and it may be difficult to determine changes in social isolation/loneliness based on treatment measures.

We focused on 38 OECD member countries because of potentially higher records of minority ethnic migration to them, which could result in fiduciary pressures on the health and care system, often negating prevention or early interventions [33]. Studies conducted in Africa and Asia, for example, where the majority share a common culture, religion, skin colour, and national interest, may have different antecedents of social isolation and loneliness than minority ethnic individuals in OECD countries. This is important to shed light on health inequalities and social factors that can impact the brain health of minority ethnic populations in those countries. Our interest in South Asian and Afro-Caribbean communities is based on a growing indication that the communities have an increased risk of dementia in the UK. Additionally, Black African/Caribbean and South Asian populations constitute among the highest migrant groups in OECD countries [34, 35].

### Search procedures

The JBI three-step approaches were utilised to search relevant articles comprehensively. This entails a preliminary limited search of select databases using text words and index items. The text words contained in the titles and abstracts of relevant articles and the index terms used to describe the articles were used to develop a full search strategy used across six databases: Medline, APA PsycINFO, Psychology and Behavioural Sciences Collection, CINAHL, Web of Science and Scopus. The support of the university librarian was sought to refine the search strategy based on concepts relating to "social isolation" OR loneliness OR "social facilitation" OR "social support" OR "social exclusion" AND "minority ethnic" OR "ethnic minority*" OR BME OR BAME OR migrant* AND intervention OR programme OR campaign.

Specific subject headings were also used for databases such as Medline and CINAHL. The search parameter included all studies published in English up until November 2023. The full search strategy is presented in the **S1 File**. In the third stage, we searched the reference list of all included articles and additional grey literature sources on websites such as AgeUK, CarersUK, Health Foundation, and SCIE.

### Data extraction and charting

Data screening was managed in Covidence software version 2.0, with detailed extraction instructions developed and agreed upon within the team for consistency. The Covidence software automatically removed duplicates and managed title and abstract screening and full-text screening. To ensure a consistent approach in line with the review questions and purpose, two reviewers (EN and AA) independently screened at least 10% of the titles and abstracts. One reviewer screened the remaining records. This step was taken to detect any systematic discrepancies that may have arisen from the reviewers and to check whether screening instructions were clear and sufficient. Early calibration and correction were done to avoid any errors. After the initial eligibility assessment, full-text screening was conducted. Two reviewers again assessed 10% of the texts independently, and any disagreements were resolved through discussion and consensus. If required, a third (SP) or fourth reviewer (KW) was consulted for further input (see S2 Table for further details). The JBI updated guidance for evidence synthesis was used to guide the data extraction and charting process. A data extraction instruction **S2 File** was developed to guide reviewers.

Data extracted from all papers included (1) study characteristics, including authors, year of publication, study design, target population, the country the study was conducted in, age and gender, (2) the intervention components, including the aims and descriptions of the

ingredients and activities of the intervention and intervention type. Other data extracted included the mode (the format of giving the intervention), dose (i.e., how many times participants were exposed to the intervention), role (who delivered the intervention and whether any cultural adaptations were applied), outcome (social isolation or loneliness), tools (scales used to assess social isolation or loneliness), barriers and facilitators (factors promoting or hindering the implementation of outcomes). We also extracted data on the theoretical underpinnings of the interventions, recording if interventions were anchored on any framework, theory, or author implicit conceptualisation from the rationale provided for why the intervention may address social isolation and loneliness. We paid attention to whether the rationale for the intervention's effectiveness was linked to brain health or social isolation and loneliness vulnerability factors.

### Data analysis

Key details of the included studies were represented in tables and graphs for easy visualisation and summarised narratively for additional clarity. Charting flow/tables included descriptive data about the study characteristics and key findings. Studies in the key finding section are grouped according to their purpose, mechanism of action and intended outcomes for easy categorisation based on a framework outlined in previous reviews [36, 37].

## Results

### Search results and study characteristics

The search strategy identified 10,479 results, of which 6,729 went through titles and abstracts screening deduplication. Of these, 29 records were eligible for full-text screening, of which 12 articles met our eligibility criteria, as shown in the Preferred Reporting Items for Systematic Reviews and Meta-Analysis (PRISMA) flow chart (**Fig 1**).

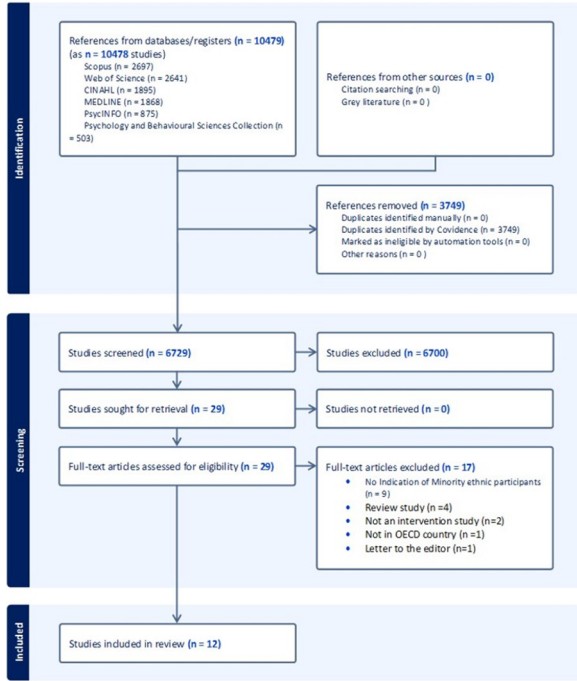

**Fig 1. PRISMA flow diagram of search result and screening process.**

The 12 studies consisted of five RCTs, six non-randomized quasi-experimental studies, and one qualitative programme. Studies were conducted in the US (67%, n = 8), Canada (25%, n = 3), or the UK (8%, n = 1) and focused on community-dwelling adults aged 60 or older (67%), with female-majority (83%) samples. Sample sizes ranged from 12 to 390 participants. Twenty-five per cent (n = 3) of the articles were published in 2020, 17% (n = 2) each in 2021 and 2015, and 8% (n = 1) each in 2022, 2012, 2010, 2006, and 2001, respectively.

Studies targeted minority ethnic populations (58%, n = 7) or mixed communities (42%, n = 5). Of 1793 participants, 60% belonged to minority ethnic populations. Among these, 45% were black or African, 18% were Asian, and 3% were Native American. For some studies, the ethnic backgrounds of the sample are often unclear and are reported as non-white (24%), other/multiracial (5%), or unreported (5%) (**Table 1**).

The studies were mostly aimed at improving outcomes related to loneliness (50%) or social connection (25%) and health behaviour/social facilitation (17%). Other variables such as depression, interest in life, social functioning, or mastery were also explored. Only one study specifically aimed to achieve social isolation and loneliness outcomes (**Fig 2**).

## Key findings

### What interventions addressing social isolation and loneliness have been designed for minority ethnic communities?

Four intervention types were categorised from the 12 included studies: social facilitation, befriending, leisure/skill development, and health education programmes (**Table 2**). Accordinging to the framework outlined in the previous review [36], the social facilitation category describes interventions that strive to engender mutual benefits for all participants by facilitating social interaction. This contrasts with befriending interventions, where volunteers usually formulate new friendships with the lonely individual. The leisure/skill development category describes interventions that tend to engage participants cognitively, physically, and socially, for example, participating in activities such as choir or exercise groups and receiving mutual support in health management. An additional category, health educational programmes, describes interventions that combine health education and related organisational and environmental supports for behaviour conducive to reducing social isolation and loneliness as well as

**Table 1. Key characteristics of included studies (n = 12).**

| Characteristics | %(N) |
|---|---|
| **Ethnicity** | |
| Black or African | 45 (476) |
| Asian | 18 (197) |
| Reported as Non-White | 24 (254) |
| Reported as Other/Multi-raced | 5 (58) |
| Native America | 3 (27) |
| Unreported | 5 (56) |
| Total categorised as **ME** | 60 (1068) |
| Total sample size | 100 (1793) |
| **Gender** | |
| Male | 16 (292) |
| Female | 82 (1464) |
| Not Known | 2 (37) |

*ME- Minority Ethnic

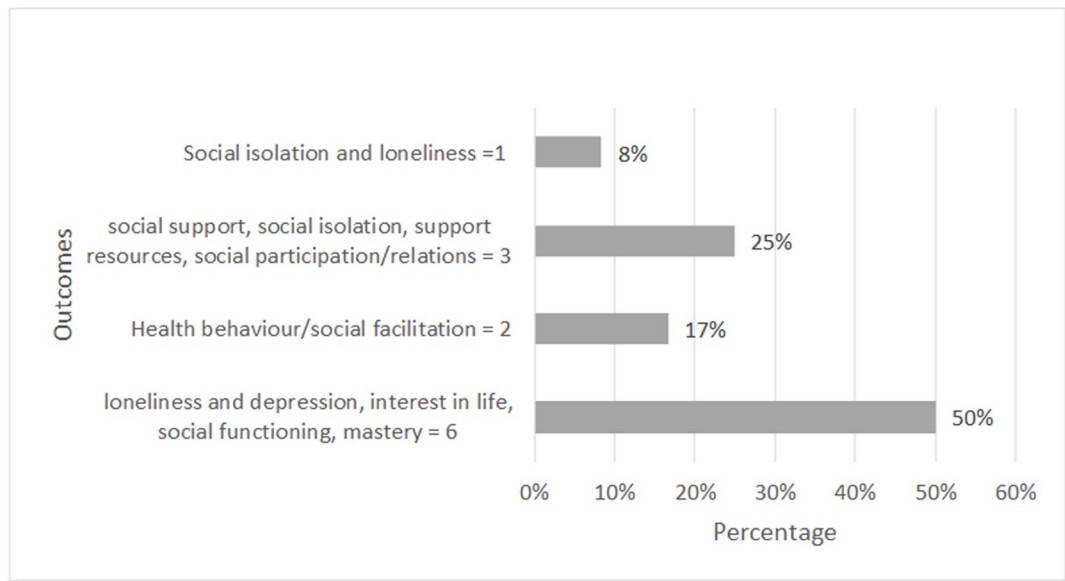

**Fig 2. Summary of outcomes measured.**

general well-being [37]. This included lectures on good nutrition and regular exercise, awareness of the prevalence of risk factors, and informal social facilitation activities such as developing friendships through disclosure, reminiscing, and discussions about current affairs.

**Social facilitation.** Of all the sources, 33% (n = 4) were included in the social facilitation category, although elements of social facilitation are evident in other categories. These studies explored the effectiveness of participant-led community building through social gathering [38], social support [39] or peer support [41] activities in refugee communities. The studies' theoretical basis, which explains why the intervention was effective, mainly focused on social integration, participation, and social belonging indicators of social isolation. The mode of delivery and intervention exposure varied according to study objectives. Studies were based on group activities [38, 39] or mixed with telephone conversations [41]. Intervention components included engaging in activities of interest, facilitating discussions to promote social relationships characterised by information exchange and peer support, immigration advice, financial counselling, coping/parenting, and sporting activities. One study, with a randomised control design, focused on addressing social difficulties, isolation, and poor access to primary care [40]. This involved one group attending facilitated activities for 10 weeks, with group activities developed in consultation with voluntary organisations for ethnic minorities. Another group was invited and offered antidepressant treatment by their GP, and the final group was invited to attend social groups as well as being offered antidepressant treatment. The idea was to compare the efficacy of social facilitation with antidepressants for women with depression to evaluate whether the combination of the two is more effective than either alone. It reported significant benefits of social facilitation over antidepressants in reducing loneliness; the difference between the social facilitation group and those who received antidepressants and attended social facilitation activities was not significant. Common among studies in this category is participants' ages, which ranged from 20 to 60 years.

All studies in the social facilitation category reported some success in reducing social isolation and/or loneliness measured through a mixed-method approach, including pre-and post-survey and interviews [38–40] or qualitative descriptions [41]. While the scale of measurement

**Table 2. Eligible studies according to intervention mechanism and type.**

| Author, year, country, Design | Intervention and control | Aim | Sample characteristics | Outcome measures | Note on effect |
|---|---|---|---|---|---|
| | | Social facilitation | | | |
| Versey et al. (2023). [38]<br><br>USA<br><br>Pre-post-test NRES | **Type:** Facilitated weekly group meeting; participants randomly assigned to attend any of 1) one meeting, 2) four meetings, or 3) six meetings<br>**Duration:** weekly event for **four months**<br><br>**No control** | To examine how well participant-led leisure time activities among a naturally occurring group encourage connectedness<br><br>**Delivery:** F2F by Community members, agency staff Reps, and academic researchers<br>**Cultural adaptation*** | **Sample size:** 12<br>**Target:** Resettled refugees<br>**Age:** 20–39 years<br>**Gender:** 100% male<br>**Ethnicity:** black = 10<br>White of Ara descent = 2 | Primary: **Social participation**<br>Secondary: **Social relationships**<br><br>**Theoretical base:** Indicators of integration framework and the social network, support, and social capital theory | The intervention helped cultivate new relationships and a sense of belonging–stronger results were observed among participants who met four or six times |
| Stewart et al. (2015) [39]<br>Canada<br><br>Mixed method (pre-post-test and interviews) | **Type:** Peer group mentoring involving like-ethnic, like-gender peers & professional mentors.<br>**Duration:** Eight groups met 1–2 hours biweekly for **7 months**.<br><br>**No control** | Evaluate the effectiveness of accessible and culturally appropriate social support identified by new refugee parents<br><br>**Delivery:** F2F group meetings co-led by peer and professional mentors from Sudan and Zimbabwe<br><br>**Theoretical base:** NI<br>**Cultural adaptation*** | **Sample size:** 85<br>**Target:** Refugee parents from Zimbabwe and Sudan<br>**Age:** 20+<br>**Gender:** male = 47, Female = 38<br>**Ethnicity:** 100% black African | **Support resources** (e.g., social, informational)<br>**Loneliness and isolation** (discrepancies between ideal and perceived interpersonal relationships)<br>**Coping** (proactive coping using the resources of others)<br>**Parenting stress** | A Wilcoxon signed-rank test did not detect a statistically significant change in loneliness (Z = 0.313, p = 0.754) or stress (Z = .792, p = .428). However, qualitative data suggested a sense of empowerment with group discussion, which helps to reduce isolation and relieve stress. |
| Gater et al. (2010). [40]<br>UK<br><br>RCT (six clusters) | **Type:** Social facilitation and psychoeducation. Participants with depression were allocated to either (a) facilitated social group meetings at the local community centre weekly over 10 weeks, (b) protocol-guided antidepressant treatment (**control**), or (c) a combination of both.<br>**Duration:** weekly sessions, follow-up = 9 months | To determine the efficacy of a social group intervention compared with antidepressants and whether the combination of the two is more productive than either alone.<br>**Delivery:** F2F group meetings by Psychiatrists, psychologists, British Pakistani mental health workers and service users | **Sample size:** 123<br>**Target:** Pakistani women with depression<br>**Age:** Mean = 41.3(11.0)<br>**Gender:**100% female<br>**100% South Asian**<br>**Theoretical base:** NI | Primary: **Changes in HRSD**<br>Secondary: **Social functioning**<br><br>**Cultural adaptation*** | There was a greater increase in social functioning in the social intervention group and the combined treatment group than in the antidepressant group at both 3 and 9 months, but these were significant only at 3 months |
| Stewart, et al. (2012) [41]<br>Canada<br><br>Qualitative (interviews) | **Type:** peer support social facilitation.<br>Eight support groups met bi-weekly for face-to-face sessions, and supplementary support via telephone was offered between sessions as participants deemed relevant.<br>**Duration:** bi-weekly 60–90 minutes and 20-minute telephone for 12 weeks<br><br>**No control** | To pilot a culturally congruent intervention that addresses the support needs of two ethnoculturally distinct refugee groups.<br>**Delivery:** Group, Peers and professional facilitators who are also of Somali and Sudanese background | **Sample size:** 58<br>**Target:** Somali and Sudanese refugees<br>**Age:** 18–54<br>**Gender:** Male = 31, Female = 27<br>**Ethnicity:** 100% black African<br>**Cultural adaptation*** | Primary: **Social exchange.**<br><br>**Theoretical base:** NI | Refugees reported increased social integration, decreased loneliness, and expanded coping repertoire following the intervention. |
| | | Befriending | | | |
| Lai et al. (2020). [42]<br>Canada<br><br>RCT | **Type:** 1:1 peer support.<br>**Intervention:** 30 Participants received two-on-one peer support services through home visits and telephone calls and attended two monthly peer support meetings.<br><br>**Control.** 30 participants only received brief telephone calls over eight weeks.<br><br>**Duration:** 10 weeks | To examine the effectiveness of the peer-based intervention on older Chinese immigrants' psychosocial well-being<br><br>**Delivery:** Individual, telephone by social workers, Immigrant volunteers of Chinese origin<br><br>**Theoretical base:** Dynamic social Impact theory<br>**Cultural adaptation*** | **Sample size:** 60<br>**Target:** Older Chinese immigrants<br>**Age** 65+<br>**Gender** Male: 22<br>Femail:38<br>**Ethnicity:** 100% Asia | **Primary:** loneliness, social support, barriers to social participation.<br>**Secondary:** depression, anxiety, happiness, life satisfaction, and purpose in life | Peer-based intervention was effective in reducing loneliness and improving resilience among socially isolated older adults |

*(Continued)*

**Table 2.** (Continued)

| Author, year, country, Design | Intervention and control | Aim | Sample characteristics | Outcome measures | Note on effect |
|---|---|---|---|---|---|
| **Kahlon, et al (2021)** [43]<br><br>USA<br><br>RCT (Individual) | **Type**: Empathetic telephone call (individual)<br>**I**: Participants received calls from 16 callers trained in empathetic conversational techniques over four weeks. Callers learned from the participants' issues and paid attention to the MOWCTX list of escalation categories, such as safety, food, or financial concerns, during a conversation to escalate for follow-up.<br>**Control**: participants received no calls | To determine whether a layperson-delivered, empathy-focused program of telephone calls could rapidly improve loneliness, depression, and anxiety in at-risk adults.<br>**Delivery**: Trained laypersons<br><br>**Duration**: Four weeks (daily for 5 days and as participants wished) | **Sample size**: 240<br>**Target**: MOWCTX clients.<br>**Age**: Mean = 69.4<br>**Gender**: 79% female<br>**Ethnicity**: Mixed.<br>Asia = 3, African American = 94<br>Unreported = 50<br>White = 93 | **Primary**: loneliness<br>**Secondary**: Depression, Anxiety, social connection<br><br>**Cultural Approach**: NI<br><br>**Theoretical base**: NI | Loneliness improved in the intervention group from M = 6.5 to 5.2 on the UCLA Scale and in the control group from 6.5 to 6.3 (df. 1.1; 95%CI, 0.5–1.1; P < .001; Cohen d of 0.48)<br>De Jong scale improved from M = 2.4 to 2.2 and no change in control group from M = 2.5 (df = 0.32:95%CI, -0.20 to 0.81; p = .06) |
| **Kotwal, et al., (2021)** [44]<br><br>USA<br><br>Mixed method NRES | **Type**: 1:1 Peer support.<br>Peers visited participants matched based on demographic and social interests (e.g., food, music, etc.) and given the flexibility to develop relationships driven by health needs and social interests.<br>**Duration**: 2 years (a median of 42 visits from peers over the first 12 months)<br><br>**No control** | To assess the effect of a peer intervention in addressing loneliness, isolation, and behavioural health needs among low-income, community-dwelling older adults of diverse racial and ethnic backgrounds.<br><br>**Delivery**: Peers through local employment agency | **Sample size**: 74<br>**Target**: Low-income older adults<br>**Age**: 59–96<br>**Gender**: Male = 58%<br>Female = 41.9%<br>**Ethnicity**:<br>White = 31<br>African = 16<br>Latino = 13<br>Native America = 3<br>Asia = 5<br>Multi-ethnic = 1 | **Primary**: loneliness, social interaction, social support<br>**Secondary**: Barrier to socialisation, depression<br><br>**Cultural adaptation***<br><br>**Theoretical base**: NI | Loneliness scores decreased by 0.8 points on average (p = 0.015), with most of the change occurring in the first 12 months, and sustained at 24 months. Interview: shared experiences were crucial in building trust, rapport, and friendships, leading to overall satisfaction |
| | | Health Education | | | |
| **Lyons and Magai (2001)** [45]<br><br>USA<br><br>RCT (group) | **Type**: Psycho-educational and social facilitation programme.<br>Intervention: participants attended a formal health education programme and participated in informal social facilitation activities.<br>**Duration**: 8 weeks, 16 weeks follow-up | To encourage good health behaviour and informal recreational component focused on participants' psychosocial wellbeing (social facilitation).<br><br>**Delivery**: healthcare and social work practitioners | Sample size: 45<br>**Target**: Older black residents of NORCs<br>**Age** = M = 73.2<br>**Gender**: 89% female<br>**Ethnicity**: 100% black America | **Primary**: health behaviour, social facilitation<br><br>Cultural adaption*<br><br>**Theoretical base**: NI | The intervention group experienced improved psychological wellbeing and sustained beyond the sessions. |
| **Collins and Benedict (2006)** [46]<br><br>USA<br><br>Pre-post NRES (Group) | **Type**: health promotion.<br>Participants attended educational programmes for 16 weeks to 1) promote health and quality by enhancing mastery, 2) create social support built around learning to decrease loneliness and stress.<br><br>**Duration**: 16 weeks (15 lessons, 32 total hours of instruction reported for each participant.<br><br>**No control** | To evaluate the effects of a community-based educational program designed to promote health by enhancing older adults' mastery while decreasing loneliness and stress<br><br>**Delivery**: co-operative extension paraprofessionals, volunteer peer educators and on-site staff | **Sample size**: 339<br>**target**: Older Adults<br>**Age**: 52–95 (M = 73.2)<br>**Gender**: 80% female<br>**Ethnicity**: Latino = 14%<br>African = 10%<br>Asian = 6%<br>White = 68% | **Primary**: Mastery, loneliness, and stress<br><br>Theoretical base: NI<br>Cultural Adaptation: NI | Participants Increased mastery from a mean score of 24.96 Â± .28 to 27.01 Â± .25 (t = 12.08, df :::::323, p < .001). Loneliness decreased from a mean score of 8.64 Â± .10 to 7.86 Â± .09 (t = -9.20, df = 329, P< .001. Stress decreased from a mean score of 16.22 Â± .34 to 13.48 Â± .32 (t = -13.51, df = 319, P< .001) |
| **Hightow-Weidman et al (2015)** [47]<br><br>USA<br><br>Pre-post NRES | **Type**: Online health/social promotion. Participants spent at least 1 hour on the HMP site weekly for four consecutive weeks to mimic time spent in face-to-face interventions and increase users' opportunities to explore multiple areas of the site.<br>**Duration**: four weeks (1-hr per week)<br>**No control** | To facilitate an online community that encourages positive norms, reflective appraisals, and supportive relationships between HIV-positive and HIV-negative YBMSM and TW.<br>**Delivery**: online, self-exploration | **Sample size**: 15<br>**Targets**: HIV-positive and HIV-negative YBMSM and TW<br>**Age**: 18–30 (M = 26.1)<br>**Gender**: YBMSM = 13, TW = 2<br>**Ethnicity**: 100% black American | **Primary**: social support (Total, emotional, tangible)<br>Secondary: depression, social isolation<br><br>**Theoretical base**: NI<br>**Cultural adaption**: NI | Despite the small sample size and limited intervention length, statistically significant improvements were seen in social support (p = .012), social isolation (p = .050), and depressive symptoms (p = .045) |

*(Continued)*

**Table 2.** (Continued)

| Author, year, country, Design | Intervention and control | Aim | Sample characteristics | Outcome measures | Note on effect |
|---|---|---|---|---|---|
| | | Leisure and Skills | Development | | |
| **May et al (2020)** [48]<br><br>USA<br><br>Pre-post NRES (Group) | **Type:** physical exercise classes. Participants enrolled in one of the four Evidence-based programs: the Arthritis Foundation Exercise Program, EnhanceFitness, Tai Chi for Arthritis, and Health Living workshop.<br>**Duration:** 3 years (variable per exercise)<br><br>**No control** | To evaluate the impact of group health classes on loneliness and social isolation in community-dwelling older adults.<br><br>**Delivery:** Professionals experienced in EBPs<br><br>**Theoretical base:** NI | **Sample size:** 382<br>**Target:** Older adults<br>**Age:** 50+ (M = 76.8)<br>**Gender:** 83.1% Female.<br>**Ethnicity:**<br>Non-Hispanic black or African = 37.93%, non-Hispanic White or Caucasian = 45.89%<br>Other = 4.59%, | **Primary:** Social Isolation and Loneliness<br><br>**Cultural adaptation:** NI | Social isolation improved—DSSI improved by 2.4% at 6 weeks compared to baseline (ER: 1.024; 95% [CI]: .010â´'1.038; p-value = 0.001), and 3.3% at 6 months (ER: 1.033; 95% CI: 1.016â´'1.050; p-value <0.001). Loneliness did not change at 6-week (ER: 0.994;5% CI: 0.962â´'1.027; p-value = 0.713), but decreased by 6.9% at 6-months (ER: 0.931; 95% CI: 0.895â´'0.968; p-value <0.001). |
| **Johnson et al. (2020)** [49]<br><br>USA<br><br>RCT (group) | **Type:** Choir Group (**physical and psychosocial**).<br>**I:** The intervention group participated in a choir programme targeting three hypothesized pathways by which a choir could promote health and well-being: cognitive, physical, and psychosocial engagement.<br>**Duration:** 44 weeks (90 minutes once a week, average attendance M = 14.4, and median of 19 of 23 sessions).<br><br>C. **waitlist controlled** | To test the effects of the Community of Voices choir intervention on the health, well-being, and healthcare costs of racial/ethnically diverse older adults<br><br>**Delivery:** Choir directors, community staff<br><br>**Theoretical base:** NI<br><br>**Cultural adaptation**\* | **Sample size:** 390<br>**Target:** Older adult<br>**Age:** 60+ (M = 71.3, sd = 7.2)<br>**Gender:** 100% Female<br>**Ethnicity:**<br>Non-white = 254(65%) | **Primary:** Psychosocial (depression, anxiety, loneliness, positive affect, and interest in life),<br>Cognitive (and Physical (lower body strength | Significant group-by-time interaction effect for loneliness (p = .02; ER = 0.34), and interest-in-life (p = .008; ER = 0.39), but the control group did not differentially change. No change in cognitive and physical outcomes |

RCT = Randomised control trial; MOWCTX = Meals on Wheels Central Texas; NI = Not indicated;

\* = presence of culturally sensitive approach; NRES = non-randomised experimental studies; YBMSM = young black men who have sex with men; TW- transgender women; F2F = face-to-face; I = intervention; C = control; UCLA = University of California Los Angeles loneliness scale; De Jong loneliness Scale; DSSI = Duke social support index

used in some studies was not statistically significant, the qualitative data provided insight into participants' experiences. For instance, the group discussions enabled participants to delve into the experiences of others and how they coped with them. This helped to empower group members, enhance social integration and alleviate loneliness and social isolation [41].

**Befriending.** Three studies(25%) in this category involved in-person and/or phone interventions. These interventions were designed for those aged 61 or older who were either homebound [43], lived independently [42] or fit the services of senior centres [44]. Unlike social facilitation interventions that aim to promote a mutually beneficial relationship, befriending interventions aim to support the lonely person [36]. The common objectives were to cultivate trusting relationships and socialisation characterised by negotiated visitation, emotional support, promotion of self-care, and problem-solving often driven by participants' health needs and social interests. In two studies, participants were randomised to intervention and control groups. The intervention group received two-to-one peer support services through home visits, telephone calls and activities such as emotional support, referrals and community resources [42] or received layperson-delivered empathetic, focused telephone calls to learn, track and escalate participants' concerns to Senior Centre agencies who followed up [43]. The control

group either received brief telephone calls over the study duration or no calls. In one study with a non-randomised design, the focus was to build trusting relationships, rapport, and friendships with peers matched to participants based on demographics and social interest [44]. Activities included providing companionship for simple errands. As the relationship grew, additional social activities such as shared meals, group outings, and walks around the city were added.

While studies reported some sustained improvement in socialisation and statistically significant decreases in loneliness and resilience in older adults, it was noted that an informal companionship approach would have a better impact than using the terms "loneliness" and "social isolation" as this was associated with stigma. Only one study [42] in this category focused solely on minority ethnic populations, with the remainder discussing mixed populations.

**Health education programmes.** Psycho-educational health interventions (25%, n = 3) were explored in three studies. The aims included a formal health education component to encourage good health, promote mastery, positive norms or reflective appraisal and an informal recreational component focusing on building psychological well-being and a supportive social network. In two studies (Mean age = 73.2) with randomised controlled [45] and non-randomised quasi-experimental design [46], participants attended a formal health education program focusing on personal safety, self-care, nutrition, exercise, productive ageing, awareness of diabetes, hypertension and cancer prevalence in participants' population and health-promotional activities that can reduce risk factors and promote good health. The second phase of psychological well-being involves the development of friendship through "a-get-to-know-each-other" activities. Classes were held once weekly for 16 weeks or four months and lasted one and a half hours. In one study, the focus was to facilitate an online community that encourages positive norms, reflective appraisals, and supportive relationships between HIV-positive and HIV-negative young black men who have sex with men (YBMSM) and transgender women (TW) [47]. YBMSM individuals experience social ostracism due to intersectional stigma [50], which can worsen when living with an HIV-positive diagnosis [51], the impact of which could lead to depression and cognitive dysfunction. The authors reported that social isolation and a lack of support were common among participants at baseline, and statistically significant improvements were found postintervention.

The psycho-educational health interventions were described as targeting mastery, health behaviour, depression, social isolation, and loneliness outcomes. Results on mastery, measured by the 7-item mastery scale [52], which assessed participants' sense of control over their lives, were reported adequate after the intervention, but loneliness and social isolation measured by the 4-item UCLA loneliness scale (version 3) were rated poorly [46]. However, the UCLA loneliness scale showed an improvement in psychological wellbeing and sustained improvement in the treatment group compared to the control group, but the health behaviour of both groups did not detect any statistically significant difference [45].

**Leisure/ skill development intervention.** Interventions in Leisure/ skill development used group exercise to improve older adults' psychosocial, cognitive, and physical wellbeing (mean age 70–76.8). In one intervention, with a pre-post-test design, participants enrolled in one of four programmes: the Arthritis Foundation Exercise Program (aimed to alleviate pains, decrease inactivity, depression and social isolation), Enhance Fitness (designed to increase strength, physical activities, and improve mood), Tai Chi for Arthritis (design to improve physical mobility, balance and psychologically healthier and healthier living workshop (aimed to offer mutual support and build confidence in participants' ability to manage their health) [48]. The study was designed to measure social isolation and loneliness, and it reported an improvement in social connectedness, as measured by the Duke Social Support Index(DSSI), and a decrease in loneliness, as measured by the UCLA Loneliness Scale. However, the study

also reported a significant loss to follow-up, as only 47.7% and 46.15% of participants completed the DSSI and UCLA scales, respectively. Meanwhile, it is unclear which elements of the intervention class influenced social connection most, given a varied approach to participants' enrolment in the exercise.

A multisite, two-armed cluster-randomised trial conducted at 12 Senior Centres targeted hypothesised pathways by which the choir could promote health and wellbeing and consisted of activities that engaged participants cognitively, physically, and socially. The intervention group started the choir immediately, and a waitlist-control group started the choir six months later [49]. Focusing on a variety of psychosocial (including loneliness, interest in life, positive affect, and sadness), physical (lower body strength, balance, and walking speed), and cognitive outcomes (memory and executive functions), the study reported no significant group-by-time differences in all the three primary outcomes. However, significant group differences were found for two of six psychosocial outcomes assessed using items from the National Institute of Health toolbox. There was a decrease in loneliness and increased interest in life for the intervention group, not the control group. Both studies were not specifically targeted to minority ethnic communities but had samples of minority ethnic populations above 10%.

## How well do the hypothesised mechanisms of action link interventions to increasing awareness of SIL risks on health factors?

**Theoretical underpinning.**    Out of 12 interventions, 83%(n = 10) did not present a clear theoretical foundation to justify the design of the intervention and their link to social isolation and loneliness indicators, including their impacts on physical and mental health. Studies often used concepts related to social isolation and loneliness(such as to increase self-efficacy and social relations [40], promote social support and coping [39, 41, 43], facilitate peer outreach [44], promote productive engagement [48] or health promotion [45] (e.g., mastery [46], physical and psychosocial engagements [49] or positive norms and caring [47] to support their rationale. However, in some papers, these rationales were not explicit, as it required a careful reading of the whole paper to dictate. In two studies with clear theoretical underpinnings [38, 42], the emphasis was on indicators of integration framework based on social domain and belonging [53] and dynamic social impact theory, based on the importance of background similarity in peer support relationships [54]. While these concepts established the culturally sensitive link to the interventions, the broader health implications of SIL are often not established. No included study was theoretically framed to account for the impacts of SIL on physical and mental well-being, including reducing dementia and other modifiable risk vulnerabilities.

## Which culturally sensitive approaches were adapted to address social isolation/loneliness factors?

Sixty-seven per cent(n = 8) indicated using culturally sensitive approaches in their interventions. The strategies were related to material, culture, accessibility, trust, partnership, and common interest (**Table 3**).

Items related to study materials (25%, n = 3) included translation of study content into participants' language or interviews being conducted in the preferred languages of the participants or using examples that reflect participants' ethnic and socioeconomic points of reference. Most common approaches related to cultural or religious significance (50%, n = 6) and targeted issues around accessibility, trust, identity, and belonging.

For example, three studies (25%) reported barriers to active involvement and participation. First, people with children found travelling through public transport an inconvenience; for

those who didn't want to bring their children, finding trusted babysitters was challenging [41]. Second, scheduling challenges were also reported, given that most participants worked multiple low-paying jobs, faced pressing issues, and had insufficient time. In some cases, participants had arrived late to the meeting and asked peer facilitators to revisit topics discussed in their absence [39, 41]. Third, participants reported that session times were not always sufficiently long to discuss complex issues such as housing problems and unemployment, and some participants were unconfident about opening up about their concerns in group sessions. One study reported resistance from family members, especially husbands, who questioned the appropriateness of the interventions and the amount of disclosure expected [40].

Authors reported strategies to encourage participation, such as using staff from the same socio-cultural backgrounds, providing culturally congruent venues, and providing childcare and transport. Other approaches identified included after-session support, co-production, and consideration of common interests between peer facilitators and participants.

The use of multilingual staff and facilitators enabled support to be provided in participants' primary languages, increasing social engagement and overcoming communication barriers [38, 40]. Similarly, providing culturally sensitive support based on local understanding reduced isolation, optimised coping, and increased belonging [39, 41]. Additionally, creating groups based on gender or ethnicity provided a safe space for open and frank discussion of concerns and ways to address them, filling gaps in knowledge and promoting social functioning and relationship building. In one paper, male participants felt their status and masculinity were being eroded, but working with peer facilitators enabled them to see how they could make significant contributions to their families and appreciate their female partners [39]. The use of participants' ethnic community centres, local radio and newspapers were primary facilitators for recruitment in two studies [40, 42].

## Which interventions are most effective in reducing social isolation and loneliness?

The variety of interventions, their target age groups, and different outcome measures and tools preclude conclusive commentary on the most effective interventions. The evidence supporting intervention effectiveness is mixed and often poor (e.g., small sample size, limited exposure to intervention activities, heterogeneity of intervention participants, lack of objective measurement, etc). Most studies had no control, which is standard for evaluating effectiveness. What is clear is that interventions aimed to reduce social isolation and loneliness among minority ethnic communities in OECD countries are still emerging. However, there is some evidence to suggest that peers and/or professionally facilitated group interventions aiming to encourage community building, social support, information exchange, and social interactions had a positive impact on participants' social functioning, cultivated a sense of belonging, reduced social isolation and loneliness, and expanded coping repertoire, particularly when they involved people or facilitators from shared ethnic networks and cultural significance [38–41]. For example, peer support group mentorship for refugee migrants enhanced support-seeking skills for coping with health-related challenges [39]. One study aimed at cultivating community building observed a more robust experience among participants who met four or six times [38]. Older adults who participated in community-based group health classes [48] or choir interventions [49] reported decreased loneliness and social isolation and increased interest in life, which are among the important factors affecting older adults. A 16-week health education intervention reported significant improvement in participants' quality of life, as participants were able to incorporate lessons-related information into their everyday lives; minority ethnic participants with lower income and higher educational levels were reported to have shown a greater

**Table 3. Identified culturally sensitive approaches used.**

| Category | Description | Examples |
|---|---|---|
| **Material** | Strategy related to intervention content and languages | 1. Translation of study material to participant language, interview conducted in participant's preferred language [41].<br>2. Descriptor and examples of study content modified according to participants' ethnic and socioeconomic point of reference; Slides, audio, and videotapes used to demonstrate appropriate health technique [45].<br>3. The choir director identified music repertoire that could be culturally tailored to each site [49]. |
| **Cultural & Religious significance** | Strategy related to participants' ethnic, cultural, and religious values | 4. Use of like-ethnic-like gender peers & professional facilitators [39, 41, 42]<br>5. Use of community choir directors peculiar to each site [49]. |
| **Accessibility** | Strategy related to equality of opportunities, barriers, and facilitators | 6. Provision of culturally appropriate venue with food; Collecting participants in taxies with a female transport facilitator; Provision of childcare [39, 40] |
| **Trust** | Strategy related to mistrust, belonging and diversity. | 7. Ensuring programme professional and administrative staff are representative of participants' ethnic & cultural backgrounds [45] |
| **Partnership** | Strategy related to design (e.g., co-production, involvement, and collaboration) | 8. Partnership model involving community members, Agency staff Reps and academic researchers [38] |
| **Common Interest** | Strategy related to person-centred approach | 9. Pairing peers and participants by language or shared life experience, participant-led approach [44] |

reduction in loneliness [46]. For one-on-one interventions, evidence suggested that having someone consistently checking in on older adults over an extended period reduced loneliness and barriers to socialisation [44]. However, based on available evidence, especially with fewer studies targeting minority ethnic populations specifically, it is still unclear which activities are most effective for reducing SIL among minority ethnic populations, requiring further research.

## Discussion

We found that Interventions designed to address social isolation and loneliness(SIL) among minority ethnic(ME) groups in OECD countries are sparse and have not been tailored to conceptually account for the modifiable impacts of SIL on the risk profiles prevalent in ME populations, including dementia risk. Given the emerging focus on early interventions and prevention, efforts will not diffuse appropriately within minority ethnic populations if intervention mechanisms are not conceptually designed to corroborate the risks of SIL on the physical and mental wellbeing of the people.

Only 7 [38–42, 45, 47] of 12 studies targeted minority ethnic populations specifically and reported effectiveness in reducing social isolation and/or loneliness. The available evidence suggests that offering group activities that enhance social participation, exchange of information resources, and peer support mechanisms, especially when they involve shared ethnic networks and cultural significance, are the most effective. Previous reviews suggested that people from ME communities value social connections within and across group members [30], and it could be that the interventions' group aspect positively influences outcomes [29]. Our review suggests that participants with a higher social exposure found it more effective than those with minimal exposure. In one study, participants inadequate exposure to intervention sessions impacted their feedback regarding the intervention's effectiveness [40], and inadequate time to explore important issues, such as employment and housing, was reported in another [41]. The post-migration social stressors often experienced by ME populations in OECD countries, including material factors (i.e., acculturation problems, access to a safe environment and employment) and interpersonal factors (i.e., exclusion, discrimination and low social status) [55, 56] may mean that they would benefit from SIL interventions expanding their knowledge

about the host culture, including how to secure employment, have good income and exchange of insight into life events. Utilising minority ethnic group associations could provide a better opportunity for people to experience sustainable satisfaction with SIL interventions. The social facilitation-focused activities in this review indicated that participants had a greater capacity to seek help and felt less isolated following the interventions, as social groups encouraged openness and trust in the exchange of informational resources on complex issues such as employment, access to services, and different coping mechanisms.

However, the mechanism of effects has been varied, often with poor operationalisation of outcomes (including a lack of objective measurement) that preclude a clear distinction of whether the aim was to reduce SIL as a primary outcome or to improve other outcomes than SIL in the lives of individuals experiencing SIL. The authors had different theories and concepts (or lack thereof) about which factors were targeted, resulting in inconsistencies in the measurement tools used. This indicates discrepancies in the mechanisms by which the interventions were designed to reduce social isolation and loneliness. Previous reviews have highlighted similar concerns [57–59], with a new guideline recommended for a more systematic approach to evaluating SIL interventions [60].

While we found that 67% of studies targeted the ageing population(60 years or above) and females, it is interesting to identify that none of the interventions was specifically theorised to promote mechanisms raising awareness of the impacts of SIL on brain health and/or dementia risk reduction. This is an interesting observation, especially when considering the fact that chronic social isolation may be higher in older adults [61] and could contribute up to 3.5% of risk factors for dementia [62]. Around the world, the number of people aged 65 and over is increasing rapidly, much so from the ME population [63]. This means that more people will rely on family carers amidst challenges in cultural caregiving expectations. While the inclusion of SIL among dementia risk factors is new, ME communities have gross inequalities in health risk awareness [64–66], suggesting the need to factor risk awareness in interventions targeting the population. Extant literature has linked SIL with various effects on brain regions, including impacts on emotions, social perceptions, cognitive functions, longevity, and an increased risk of neurogenerative diseases [67–70]. Co-producing SIL interventions with ME associations will enhance greater knowledge of brain health and enhance sustainable outcomes.

Additionally, SIL interventions through informal ME associations could be combined with health education to ensure better outcomes. Health education interventions in our review were reported effective due to participants being able to incorporate lesson-related information into their lives on an everyday basis, converting abstract concepts into practical applications [45, 46]. Aside from the social benefits of coming together, health promotion interventions are an integral approach to raising awareness of disease prevalence, prevention and management across populations [71] and have been shown to be effective in improving knowledge and lifestyle changes in a range of physical health issues [72]. This review suggests that culturally appropriate strategies related to ethnic and religious values seemed to have facilitated trust, retention and participation in the research, which aligns with the principle for adapting behavioural Interventions for minority ethnic populations [73]. However, most adaption did not recognise cultural differences among participants; about 33% (n = 4) did not indicate adopting culturally appropriate measures. Previous studies have shown that culturally tailored approaches improved healthcare outcomes, reduced barriers to treatment, and addressed health disparities for minority ethnic groups [74], with shared-identity support groups having overall positive impacts on certain dimensions of loneliness among migrants and minority ethnic groups [30].

In this review, minority ethnic community centres were utilised for participant recruitment in only two studies. Leveraging informal community resources and settings that are familiar

and accessible to minority ethnic groups has had a greater impact on the uptake of health messages [75–77]. In light of the concept of the social-ecological model [78], individuals are a microcosm of larger social systems by which interactive characteristics between individual and their environments underlie health outcomes. Equitable Brain health and dementia risk reduction campaigns must involve different formal and informal minority ethnic associations to be sustainable. The concept of race and space and the phenomenon of "gentrification" [79] have marked and shaped the lived experiences of minority ethnic populations in OECD countries; exploring the role of ethnic minority resource centres in SIL interventions may identify more sustainable approaches to reducing SIL in the communities. Arguments based on important elements of social capital, such as the ability to trust and work with others, deprivations and capabilities, and having a broad social network, are likely to have significant consequences for social isolation and cohesion in OECD countries [80, 81]. These factors can determine how migrants and ethnic minorities contribute to either fostering or hindering social cohesion, which can affect their overall wellbeing. Further research is needed to identify the best approaches for reducing social isolation in diverse ethnic communities.

## Limitations

The review included only those studies conducted in OECD member states and published in English. We did not include interventions in primary care settings, such as hospitals, rehabilitation centres, care homes or nursing homes, given that such facilities already implement physical and psychological support that may be difficult to separate from intervention treatment outcomes. Therefore, there could be other studies on reducing social isolation in ethnic minority populations that were not considered in this review because they were either in other languages or conducted outside of the member states or settings. Moreover, we did not find any grey literature, such as non-peer-reviewed papers, thesis, or dissertations, which could have widened the number of papers found. We addressed this limitation by ensuring a transparent, comprehensive search and standardised data extraction procedures involving three independent reviewers. Most studies were conducted in the US and Canada and may not be generalised due to different designations of ethnic demography across OECD countries. Scoping reviews have certain limitations that are worth noting. One such limitation is that there is no formal evaluation of the quality of the evidence presented in the included studies. This means there is no assessment of effect sizes, statistical power, or a meta-evaluation of results that meet a high threshold of statistical criteria.

## Conclusion

This review found that interventions aimed at tackling social isolation and loneliness among minority ethnic groups in OECD countries are scarce and have not considered the relationship between vulnerability factors for social isolation/loneliness and the modifiable risk profile of ME populations. Studies have largely targeted individual capabilities, leaving a gap in understanding the interactions between individuals and their social, physical, and cultural environments. This discovery highlights the challenges in promoting positive lifestyle behaviours as protective factors for brain health and other physical health conditions, particularly considering the projected rise of dementia in minority ethnic populations in the UK. More research is needed to improve our theoretical understanding of the links between SIL and modifiable risk factors for neurogenerative and cardiovascular conditions. Studies should concentrate on collaborating with minority ethnic communities to identify common strategies that can improve social well-being, especially in the UK, where only one study published in 2006 was identified. This will help better understand how communities and societal factors affect social isolation

and loneliness among the population. Such a partnership would help identify differences between and within ethnic minorities and ensure that the implementation efforts and outcomes are sustainable beyond individual capabilities to address the issues of social isolation and loneliness.

## Supporting information

**S1 Table. Preferred Reporting Items for Systematic Reviews and Meta-Analyses extension for Scoping Reviews (PRISMA-ScR) checklist.**
(PDF)

**S2 Table. Additional information on included studies and extraction decisions.**
(PDF)

**S1 File. Search strategy in CINAHL.**
(PDF)

**S2 File. Data extraction form.**
(PDF)

**S3 File.**
(PDF)

## Author Contributions

**Conceptualization:** Emmanuel Sunday Nwofe, Amirah Akhtar, Sahdia Parveen, Karen Windle.

**Data curation:** Emmanuel Sunday Nwofe.

**Formal analysis:** Emmanuel Sunday Nwofe.

**Investigation:** Emmanuel Sunday Nwofe, Amirah Akhtar, Sahdia Parveen.

**Methodology:** Emmanuel Sunday Nwofe.

**Project administration:** Emmanuel Sunday Nwofe.

**Supervision:** Sahdia Parveen, Karen Windle.

**Validation:** Emmanuel Sunday Nwofe, Amirah Akhtar, Sahdia Parveen, Karen Windle.

**Visualization:** Emmanuel Sunday Nwofe.

**Writing – original draft:** Emmanuel Sunday Nwofe.

**Writing – review & editing:** Emmanuel Sunday Nwofe, Amirah Akhtar, Sahdia Parveen, Karen Windle.

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
