## [Decision Letter · Decision Letter 0]

28 Jun 2024

PONE-D-24-19431Dementia Risk Factors and Brain Health: A Scoping Review of Interventions to Reduce Social Isolation and Loneliness in Minority Ethnic PopulationsPLOS ONE

Dear Dr. Nwofe,

Thank you for submitting your manuscript to PLOS ONE. After careful consideration, we feel that it has merit but does not fully meet PLOS ONE’s publication criteria as it currently stands. Therefore, we invite you to submit a revised version of the manuscript that addresses the points raised during the review process.

**The reviewers raised several critical points which should be addressed in the revision process. Please find their comments. **

We look forward to receiving your revised manuscript.

Kind regards,

Daichi Sone

Academic Editor

PLOS ONE

Journal Requirements:

2. Please amend either the abstract on the online submission form (via Edit Submission) or the abstract in the manuscript so that they are identical.

Reviewers' comments:

Reviewer's Responses to Questions

**Comments to the Author**

1. Is the manuscript technically sound, and do the data support the conclusions?

Reviewer #1: No

Reviewer #2: Yes

2. Has the statistical analysis been performed appropriately and rigorously? 

Reviewer #1: N/A

Reviewer #2: Yes

3. Have the authors made all data underlying the findings in their manuscript fully available?

Reviewer #1: Yes

Reviewer #2: Yes

4. Is the manuscript presented in an intelligible fashion and written in standard English?

Reviewer #1: Yes

Reviewer #2: Yes

5. Review Comments to the Author

**Reviewer #1: **The viewpoint is excellent, but there are various problems with the logic and details.

This paper reviews studies of interventions for ethnic minority groups on social isolation and loneliness in the community. This viewpoint is important.

The participants in the studies seem to be mainly non older people. The authors, sorry if my understanding is not clear enough, connect the results to the dementia prevention. This is a logical leap. I recommend a major revision to simply the scoping review of interventions in the community against SIL. If the authors absolutely insist on discussing dementia prevention, the authors should do that in the discussion section.

The paper is also longer than it should be. For example, in the results section, the authors mention the scale used, however this is in the table and is not essential and should be removed.

Below are the details

-Abstract

There are repetitions, i.e., “However, despite numerous studies on reducing social isolation and loneliness in the general population, not much is known about interventions aimed at reducing these factors in minority ethnic populations as part of dementia risk reduction efforts”

-References.

52 and 62 are the same references. This was very confusing to the reader.

**Reviewer #2:** The authors focused on social isolation and loneliness of minority ethnic populations and aimed to review the interventions for these subjects in the contect of brain health and dementia risk reduction efforts.

This review paper is very well designed and well written. However, as a result, no included study was framed in the context of brain health or dementia prevention. In this long article, there is no result on dementia risk factors and brain health.

This result is inevitable, however, the authors need to change the article title and Background section. (For example, short title "Dementia Risk Factors and Brain Health” is inappropriate.)

I recommend the authors to exclude the context of Dementia Risk Factors and Brain Health and to review on interventions to reduce social isolation and loneliness in minority ethnic populations.

6. PLOS authors have the option to publish the peer review history of their article (what does this mean?). If published, this will include your full peer review and any attached files.

Reviewer #1: No

Reviewer #2: No

---

## [Author Response · Author response to Decision Letter 0]

19 Jul 2024

Journal Requirements

Comments from the Editor

Please ensure that your manuscript meets PLOS ONE's style requirements, including those for file naming

The manuscript has been revised following PLOS ONE’s style guidelines, including file naming (see pages 1, 9 and 11).

Abstracts

Comments from the Editor

Please amend either the abstract on the online submission form (via Edit Submission) or the abstract in the manuscript so that they are identical. 

The abstract on the online submission form is now amended to mirror the abstract within the manuscript (see pages 1-2).

Supporting Information file

Comments from the Editor

Please include captions for your Supporting Information files at the end of your manuscript and update any in-text citations to match accordingly.

The supporting information files have now been updated with relevant captions and in-text citations (see pages 5, 7-8 for in-text citations and page 41 for captions).

Reviewer 1: Logic and details

I recommend a major revision to simply the scoping review of interventions in the community against SIL. If the authors absolutely insist on discussing dementia prevention, the authors should do that in the discussion section.

We have revised the manuscript to focus on those interventions that mitigate social isolation and loneliness in the community. We have changed the title of the paper to reflect this and refined the questions, the abstracts, and the introduction (see pages 1-4). 

While we wanted to explore protective factors for brain health, focusing on those interventions which negate social isolation and loneliness among minority ethnic populations, we found no papers that incorporated such discussion or focus. Given the importance of social isolation and loneliness as risk factors for dementia, we have brought this element together in the discussion section (see pages 31-32). 

Reviewer 1: The paper is also longer than it should be. For example, in the results section, the authors mention the scale used, however this is in the table and is not essential and should be removed.

We have removed the scale measurement in the table and reduced the introduction to ensure that we are within the necessary word count.

Reviewer 1: Abstract

There are repetitions, i.e., “However, despite numerous studies on reducing social isolation and loneliness in the general population, not much is known about interventions aimed at reducing these factors in minority ethnic populations as part of dementia risk reduction efforts.”

All repetitions have now been corrected and revised.

Reviewer 1: References.

52 and 62 are the same references. This was very confusing to the reader.

These have now been corrected with the WHO (2014) referenced at #52 and, Hynie (2018) referenced at #62.

Reviewer 2: Title and context

No included study was framed in the context of brain health or dementia prevention. In this long article, there is no result on dementia risk factors and brain health. This result is inevitable, however, the authors need to change the article title and Background section. (For example, short title "Dementia Risk Factors and Brain Health” is inappropriate.)

We have revised the title of this paper to exclude brain health and dementia risk factors. This now reads: Interventions to reduce social isolation and loneliness among minority ethnic populations in OECD Countries: A scoping review. 

Reviewer 2: I recommend the authors to exclude the context of Dementia Risk Factors and Brain Health and to review on interventions to reduce social isolation and loneliness in minority ethnic populations.

As we have previously highlighted (see Reviewer 1 logic and details), we have revised the manuscript to bring brain health into the discussion section.

---

## [Decision Letter · Decision Letter 1]

29 Jul 2024

PONE-D-24-19431R1Interventions to reduce social isolation and loneliness among minority ethnic populations in OECD Countries: A scoping reviewPLOS ONE

Dear Dr. Nwofe,

Thank you for submitting your manuscript to PLOS ONE. After careful consideration, we feel that it has merit but does not fully meet PLOS ONE’s publication criteria as it currently stands. Therefore, we invite you to submit a revised version of the manuscript that addresses the points raised during the review process.

Thank you for the revision process to address the reviewers' comments. One reviewer proposed some minor issues below. Please consider them. 

We look forward to receiving your revised manuscript.

Kind regards,

Daichi Sone

Academic Editor

PLOS ONE

Journal Requirements:

Reviewers' comments:

Reviewer's Responses to Questions

**Comments to the Author**

1. If the authors have adequately addressed your comments raised in a previous round of review and you feel that this manuscript is now acceptable for publication, you may indicate that here to bypass the “Comments to the Author” section, enter your conflict of interest statement in the “Confidential to Editor” section, and submit your "Accept" recommendation.

Reviewer #1: All comments have been addressed

Reviewer #2: All comments have been addressed

2. Is the manuscript technically sound, and do the data support the conclusions?

Reviewer #1: Yes

Reviewer #2: Yes

3. Has the statistical analysis been performed appropriately and rigorously? 

Reviewer #1: N/A

Reviewer #2: Yes

4. Have the authors made all data underlying the findings in their manuscript fully available?

Reviewer #1: Yes

Reviewer #2: Yes

5. Is the manuscript presented in an intelligible fashion and written in standard English?

Reviewer #1: Yes

Reviewer #2: Yes

6. Review Comments to the Author

Reviewer #1: The revised manuscript is very well refined, informative, and discoverable. If authors could revise a few more points, I would suggest acceptance.

The finding that the interventions were divided into four categories is a very important finding, but it is not clearly mentioned in the abstract. Also, it is unbalanced to emphasize that group activities are superior because, after all, at the current level of evidence, it is impossible to say which intervention is superior. The priority should be written in favor of the need for future research.

Also, in the second paragraph of the introduction, dementia is suddenly mentioned in detail, but it is not consistent with the whole picture. Perhaps authors do not want to delete it because it relates to the initial passion of this project, but introductions could be described in a simpler manner.

From a medical researcher's perspective, the results section could also be shorter. It is not necessary for the reader to know detail about scale. Shorter would increase readability, but I would leave this to the editor to decide.

Reviewer #2: My concern have been corrected. The revisions have made this paper more organised in its argument and easier to read.

7. PLOS authors have the option to publish the peer review history of their article (what does this mean?). If published, this will include your full peer review and any attached files.

Reviewer #1: No

Reviewer #2: No

---

## [Author Response · Author response to Decision Letter 1]

30 Jul 2024

Journal Requirements

Comments from the Editor

The reference list has been reviewed, and incorrect references have been corrected or removed. For example, 

In Reference number 2, WHO is now fully written as World Health Organisation. 

Reference number 34 was removed and replaced with United Nations Department of Economic and Social Affairs Population division. World Migration in Figures. OECD-UNDESA; 2013

Reference 35 is now replaced with McAuliffe M, Triandafyllidou A. Word migration report 2022. 2021.

Reference 69 is now reference 66 and have been corrected from Ageing cfpo to Centre for planning on Ageing. The future ageing of the ethnic minority population of England and walesWales. 2020. Available from http://www.cpa.org.uk/BMEprojections/BMEprojections.html

Reference 77 is now 74 and WHO have been written in full.

Reviewer 1: Abstract

The finding that the interventions were divided into four categories is a very important finding, but it is not clearly mentioned in the abstract.

We have revised and included this in the abstract. 

Reviewer 1: Findings

Also, it is unbalanced to emphasise that group activities are superior because, after all, at the current level of evidence, it is impossible to say which intervention is superior. The priority should be written in favor of the need for future research.

We have revised the manuscript to indicate that the available evidence makes it difficult to determine which intervention was superior and recommended further research. (See the abstract and page 28 of the manuscript). 

Reviewer1: Introduction

 Also, in the second paragraph of the introduction, dementia is suddenly mentioned in detail, but it is not consistent with the whole picture. Perhaps authors do not want to delete it because it relates to the initial passion of this project, but introductions could be described in a simpler manner.

We have revised the paragraph, identifying dementia as one amongst those other risks associated with social isolation and loneliness (see page 3 of the manuscript). 

Reviewer 1: Result

From a medical researcher's perspective, the results section could also be shorter. It is not necessary for the reader to know detail about scale. Shorter would increase readability, but I would leave this to the editor to decide.

We have reviewed the result section and removed discussion around the different measurement scales applied across the studies, except where it formed part of the narrative of effectiveness (see pages 20-23 of the Revised manuscript with track changes or 19-23 of the Manuscript).

---

## [Editor Report · Decision Letter 2]

5 Aug 2024

Interventions to reduce social isolation and loneliness among minority ethnic populations in OECD Countries: A scoping review

PONE-D-24-19431R2

Dear Dr. Nwofe,

We’re pleased to inform you that your manuscript has been judged scientifically suitable for publication and will be formally accepted for publication once it meets all outstanding technical requirements.

Kind regards,

Daichi Sone

Academic Editor

PLOS ONE
---

## [Editor Report · Acceptance letter]

16 Aug 2024

PONE-D-24-19431R2 

PLOS ONE

Dear Dr. Nwofe, 

I'm pleased to inform you that your manuscript has been deemed suitable for publication in PLOS ONE. Congratulations! Your manuscript is now being handed over to our production team.

Kind regards, 

on behalf of

Dr. Daichi Sone 

Academic Editor

PLOS ONE